# Biochar-Induced Mitigation Potential of Greenhouse Gas Emissions Was Enhanced under High Soil Nitrogen Availability in Intensively-Irrigated Vegetable Cropping Systems

Yunfeng Zhang [1], Delight Hwarari [2], Yuwen Yang [3], Ailing Huo [1], Jinyan Wang [3,*] and Liming Yang [2,*]

[1] Jiangsu Key Laboratory for Eco-Agricultural Biotechnology around Hongze Lake, School of Life Science, Huaiyin Normal University, Huai'an 223001, China
[2] College of Biology and the Environment, Nanjing Forestry University, Nanjing 210037, China
[3] Excellence and Innovation Center, Jiangsu Academy of Agricultural Sciences, Nanjing 210014, China
[*] Correspondence: wangjy@jaas.ac.cn (J.W.); yangliming@njfu.edu.cn (L.Y.)

**Abstract:** Intensive irrigation coupled with excessive nitrogen (N) fertilizer input has resulted in high soil greenhouse gas (GHG) emissions in vegetable cropping systems. Biochar as a soil amendment has been advocated as a desirable option to reduce GHG emissions in agricultural systems, but its interactive effects with soil N availability in vegetable systems have yet to be clarified. We performed a field study to examine how biochar interacts with N fertilizer in driving annual methane ($CH_4$) and nitrous oxide ($N_2O$) emissions from an intensively-irrigated greenhouse vegetable cropping system acting as both sources of atmospheric $CH_4$ and $N_2O$ in subtropical China. Biochar amendment significantly increased soil $CH_4$ emissions by 33% and 85%, while it decreased soil $N_2O$ emissions by 22% and 12% with and without N fertilizer input, respectively. Fertilizer N combination weakened the positive response of $CH_4$ to biochar while it enhanced the mitigation potential of biochar for $N_2O$. Annual direct emission factors of fertilizer N for $N_2O$ were estimated to be 1.35% and 1.94% for the fields with and without biochar amendment, respectively. Annual flux-sustained global warming potential (SGWP) and greenhouse gas intensity (GHGI) were significantly decreased by biochar amendment, and this mitigation effect was enhanced with fertilizer N combination. Altogether, we highlight that biochar can reconcile higher yield and lower climatic impact in intensive vegetable cropping systems in subtropical China, particularly in vegetable soils with high N availability.

**Keywords:** $CH_4$; $N_2O$; SGWP; GHGI; vegetable cropping system

## 1. Introduction

Atmospheric methane ($CH_4$) and nitrous oxide ($N_2O$) are two long-lived greenhouse gases (GHGs) with flux-sustained global warming potentials (SGWP) of 45 and 270 times that of carbon dioxide ($CO_2$), respectively over a 100-year time horizon [1]. Agriculture is a major global source of $CH_4$ and $N_2O$ to the atmosphere, mainly contributed by flooded rice production, fertilizer nitrogen (N) application and livestock production [2–4]. However, large uncertainties currently still exist on the direction and magnitude of GHGs emissions across various agroecosystems, since different agroecosystems often experience significant differences in management practices (e.g., water regime, fertilizer application) [5–7].

Vegetable fields, as one of the most human-impacted agroecosystems in China, are undergoing excessive N fertilization, frequent irrigation and multiple cropping rotations in a single year [8–10]. The vegetable production area in China represents around 45% of the global total [11]. In 2014, the well-controlled greenhouse vegetable cropping area expanded, and accounted for 17.6% of the national total vegetable planting area [12]. Greenhouse vegetable cropping systems have lower N use efficiency yet receive greater N fertilization rates, which are nearly 3–5 times greater than those in other upland cropping systems [13,14]. Presumably, this excessive N load in addition to frequent irrigation could

stimulate both soil $CH_4$ and $N_2O$ emissions from vegetable cropping systems [10,15,16], although $CH_4$ emissions from vegetable soils are poorly represented.

Biochar addition to agricultural soils has provided a desirable manipulative option to sustain crop productivity while reducing GHGs release [10,17,18]. A recent global synthesis research recorded an average of 54% mitigation of $N_2O$ emissions following biochar amendment, depending on specific soil features and fertilizer N sources [19]. In contrast, according to the latest synthesized finding by Ji et al. [20], the role of biochar in mitigating soil $CH_4$ emissions might have been largely realized in individual studies, especially in pot or laboratory-controlled environments, but showing no pronounced effect in field-based studies. However, less evidence has been documented on biochar-stimulated soil $CH_4$ emissions [21,22]. To date, numerous studies have dedicated to examining biochar effects on soil $CH_4$ and $N_2O$ fluxes mostly in traditional grain cultivation systems (e.g., rice, maize or wheat), but highly expected in vegetable production systems [10,15]. Furthermore, little is known regarding how biochar interacts with N fertilizer to influence the net flux-sustained global warming potential and greenhouse gas intensity (GHGI) of $CH_4$ and $N_2O$ in these highly-irrigated lowland vegetable cropping systems in subtropical China [10]. Generally, biochar amendment leads to a trade-off between $CH_4$ and $N_2O$ emissions, however, this trade-off effect may depend on soil N availability.

Given the extent of biochar to mitigate soil GHGs emissions from agroecosystems, we examined how biochar interacts with N fertilizer to regulate soil $CH_4$ and $N_2O$ fluxes in an frequently-irrigated vegetable rotation in subtropical China. The main goal was to quantify the independent and combined effect of biochar amendment with N fertilizer input on soil $CH_4$ and $N_2O$ fluxes over a vegetable annual rotation. Since the vegetable annual rotation consisting of capsicum (*Capsicum annuum*), tomato (*Solanum lycopersicum*) and Chinese cabbage (*Brassica chinensis*) have been recognized as important sources of $N_2O$ and $CH_4$. Therefore, $N_2O$ and $CH_4$ fluxes and their measurements recorded over a whole vegetable annual rotation cycle will provided insight on the annual direct greenhouse gasses emissions. We hypothesized that the mitigation effect of biochar on $N_2O$ emissions could be altered by the extent of soil N enrichment in vegetable fields.

## 2. Materials and Methods

### 2.1. Site Description and Biochar Processing

The field experiment was established over the period from 1 September 2019 to 10 August 2020 in the Hongze Lake agricultural experimental center at suburban Huai'an, Jiangsu Province, China (33°24′ N, 119°10′ E). The site has a silt loam soil (0–20 cm), with a bulk density of 1.2 g $cm^{-3}$, soil pH of 5.9, and total carbon and nitrogen contents of 16.1 g $kg^{-1}$ and 2.23 g $kg^{-1}$, respectively. The initial extractable $NH_4^+$ and $NO_3^-$ averaged 27.8 mg N $kg^{-1}$ and 168.9 mg N $kg^{-1}$ prior to the start of the experiment, respectively.

The biochar used in this study is a by-product of wheat straw charcoal production subject to fast pyrolysis at 350–550 °C. It has a pH of 10.9, and total carbon and nitrogen contents of 467.2 g $kg^{-1}$ and 6.0 g $kg^{-1}$, respectively, but negligible extractable $NH_4^+$ and $NO_3^-$. Prior to the field experiment, biochar was thoroughly mixed with the 0–20 cm surface soil before vegetable seedlings were transplanted.

### 2.2. Field Experiment

A completely randomized block design consisting of four field treatments with six replicates was adopted in this study: (1) Control (CK): without N fertilizer and biochar application; (2) Conventional N fertilizer treatment (N): according to the local fertilizer practice; (3) Biochar treatment (B): the biochar treatment did not receive any N-fertilizer (accept the biochar); (4) N fertilizer together with biochar application (B+N). The treatment plots were randomly set in the greenhouse, with an area of 3.2 × 3.2 $m^2$, and a local typical vegetable cropping rotation was implemented in this study. Seeds of Chinese cabbage (*Brassica chinensis*) were directly sown into the experimental plots, and the seedlings of cap-

sicum (*Capsicum annuum*) and tomato (*Solanum lycopersicumes*) were grown in the seedling nursery for about one month and then transplanted into the experimental plots.

All experimental plots were plowed and leveled off before each cropping event occurred and basal fertilizer was applied once prior to new vegetable crop settlement. To be a representative of the local practice, compound fertilizer (N:P$_2$O$_5$:K$_2$O = 15%:15%:15%) was selected as N fertilizer throughout the annual cropping rotation. Basal fertilizer was spread over the field and then mixed thoroughly with the soil by plowing on 27 August 2019 during capsicum season, on 1 February 2020 during tomato season, and on 10 June 2020 during Chinese cabbage season (Table 1). For top-dressing events, compound fertilizer was applied accompanied with irrigation water; For biochar-amended plots, biochar was applied at a rate of 40 t ha$^{-1}$ and thoroughly integrated with the top soil at a depth of 0–20 cm prior to the field experiment. For each vegetable cropping season, an automatic irrigation system was equipped to guarantee irrigation frequency of 3–4 times per week. Apart from biochar amendment, all field management practices were consistent with local vegetable production. The detailed information on the cultivation and fertilization is shown in Table 1.

**Table 1.** Field managements, N fertilization events and rates over the annual vegetable cropping rotation.

| Cropping System | | | N Fertilizer Application | | |
|---|---|---|---|---|---|
| Crop Phase | Sowing/Transplanting | Harvest | Date | Event | Rate (kg N ha$^{-1}$) |
| Capsicum [a] | 1 Sept. 2019 | 2 Dec. 2019 | 27 Aug. 2019 | Basal | 150 |
| | | | 28 Sep. 2019 | Top dressing | 75 |
| | | | 1 Nov. 2019 | Top dressing | 75 |
| Tomato [a] | 4 Feb. 2020 | 18 May 2020 | 1 Feb. 2020 | Basal | 120 |
| | | | 9 Apr. 2020 | Top dressing | 90 |
| Chinese cabbage [b] | 15 Jun. 2020 | 10 Aug. 2020 | 10 Jun. 2020 | Basal | 150 |

[a] Seedlings initially grown in nursery and then transplanted to the fields. [b] Seeds directly sown in the fields.

### 2.3. CH$_4$ and N$_2$O Flux Measurements

Greenhouse gas fluxes were determined in situ by the static chamber approach. Specially-made sampling collars were initially inserted into the soil (at 0.15 m soil depth) and were placed in each plot with no crop growth presence to guarantee periodic placement of gas sampling chambers during the observation period. A well-shaped groove (5 cm in depth) at the top of each sampling collar was designed to seal the rim of the chamber with water when sampling. The sampling chambers were made of opaque PVC materials with a size of 50 (length) × 50 (width) × 50 (height) cm, and a fan was equipped inside to allow for complete gas mixing during gas sample collecting. In order to avoid drastic temperature changes inside the chamber during the sampling period, we wrapped each chamber with a layer of sponge and aluminum foil. Soil CH$_4$ and N$_2$O fluxes were generally measured twice a week, and daily or every two days following N fertilizer application and irrigation for one week. Gas samples were taken at 0, 5, 10, 15, 20 min after the chamber was placed on the PVC frames between 0830 and 1030 hrs local standard time at each sampling date. Individual gas samples of 60 ml were collected from the headspace of the chamber using gas sampling bags and were moved to the laboratory for analysis by GC (Agilent 7890A Gas Chromotography, Kansas City, KS, USA) within a few hours. Methane and N$_2$O fluxes were determined using a non-linear approach [23]. Seasonal or annual total of CH$_4$ and N$_2$O emissions were obtained by accumulating the release rates between every two adjacent intervals of measurements.

### 2.4. Calculation of SGWP and GHGI

To provide an insight into the attribution of greenhouse vegetable cropping systems to climate change as influenced by N fertilizer input and biochar amendment, we estimated the combined flux-sustained global warming potential (SGWP) of CH$_4$ and N$_2$O emissions

over the annual vegetable rotation system. According to the latest weight metrics, the warming forces of $CH_4$ and $N_2O$ are 45 and 270 times that of $CO_2$, respectively [1].

$$\text{SGWP (t } CO_2\text{-equivalent ha}^{-1}) = 45 \times CH_4 \text{ (kg ha}^{-1}) + 270 \times N_2O \text{ (kg ha}^{-1}) \quad (1)$$

The greenhouse gas intensity (GHGI) linking SGWP to ecosystem production was then determined by dividing the net SGWP by the fresh vegetable yields [10]

$$\text{GHGI (t } CO_2\text{-equivalent t}^{-1}) = \text{SGWP/vegetable fresh yield} \quad (2)$$

Vegetable yields were determined following each vegetable-growing season by gathering the weights of all above ground vegetable components for each plot.

*2.5. Other Data Measurements*

Soil samples were collected and mixed thoroughly prior to vegetable cultivation to measure topsoil physicochemical properties for each field plot. Soil temperature and moisture (0–20 cm) for each treatment plot were measured at each gas sampling using a digital thermometer and a portable rod probe (MPM-160; Armidale NSW 2350, Australia), respectively. The measured soil moisture was further transformed into water-filled pore space (WFPS) by the following formula:

$$\text{WFPS = (soil volumetric water content/(1 } - \text{ (soil bulk density/2.65)}) \times 100\%) \quad (3)$$

where 2.65 Mg m$^{-3}$ was the assumed soil particle density.

Soil samples at 0–20 cm depth were collected for determining soil $NO_3^-$-N and $NH_4^+$-N contents every 10 to 15 days. Soil bulk density (volumetric density = property of powders/solids) was measured using the cutting ring approach [24].

*2.6. Statistical Analyses*

Results were presented as means with standard errors (Mean $\pm$ SE, *n*=6). Differences in seasonal or annual cumulative $CH_4$ and $N_2O$ emissions as affected by N fertilizer, biochar and their interaction were examined by a two-way analysis of variance (ANOVA). Linear or non-linear regressions were performed to examine the relationships of soil $CH_4$ and $N_2O$ fluxes with soil temperature, moisture (WFPS) and mineral N ($NH_4^+$+$NO_3^-$). All statistical analyses were completed using SPSS statistics version 21 (SAS Institute, Cary, NC, USA, 2019).

**3. Results**

*3.1. Soil $CH_4$ Fluxes*

Soil $CH_4$ fluxes showed similar patterns across treatments over the annual vegetable cultivation cycle, independent of N fertilization and biochar addition (Figure 1b). The greenhouse vegetable cropping system generally acted as a minor source of atmospheric $CH_4$. Soil $CH_4$ fluxes averaged 0.15 mg m$^{-2}$ h$^{-1}$ and 0.12 mg m$^{-2}$ h$^{-1}$ for the N fertilized treatments with and without biochar amendment, respectively. Annual total of soil $CH_4$ emissions was significantly increased by N fertilization (192–306%), and was most facilitated in the capsicum growing season (Tables 2 and 3). Compared with the control, biochar amendment increased $CH_4$ emissions by 85% and 33% for the biochar alone and combined treatments, respectively. The largest $CH_4$ emission rate occurred in the combined treatment, which was nearly five-fold that of the control (Figure 1c).

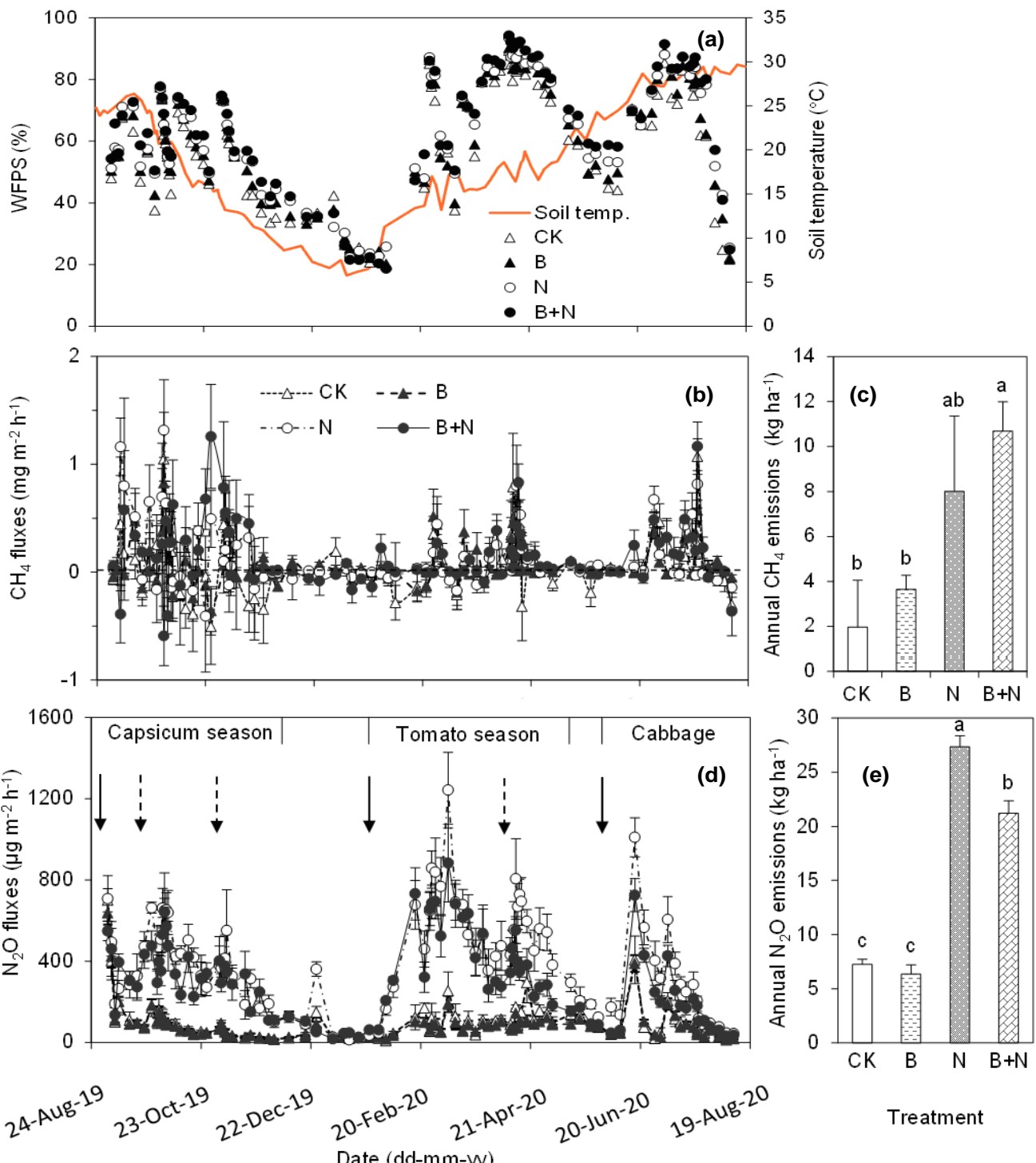

**Figure 1.** Seasonal dynamics of soil temperature and WFPS (**a**), CH$_4$ fluxes (**b**), Annual CH$_4$ emisions (**c**), N$_2$O fluxes (**d**) and Annual N$_2$O emissions (**e**) across treatments from the greenhouse vegetable cropping system over the 2019–2020 annual cycle. CK represents the Control, no N fertilizer and biochar application; B, biochar alone; N, N fertilizer alone; B+N, N fertilizer and biochar amendment. Solid and dotted arrows indicate basal and topdressing events, respectively.

**Table 2.** Seasonal and annual total of $CH_4$ and $N_2O$ emissions and fertilizer N-induced $N_2O$ emission factors across the annual cropping systems.

| Cropping Phase | $CH_4$ Emissions (kg $CH_4$ ha$^{-1}$) | | | | $N_2O$ Emissions (kg $N_2O$-N ha$^{-1}$) | | | | Emission Factor of $N_2O$ (%) | |
|---|---|---|---|---|---|---|---|---|---|---|
| | Control | Biochar (B) | N Fertilizer (N) | B+N | Control | Biochar (B) | N Fertilizer (N) | B+N | N Fertilizer (N) | B+N |
| Capsicum | 0.04 ± 1.96 c | 0.86 ± 0.53 b | 4.17 ± 2.86 a | 5.93 ± 1.73 a | 1.96 ± 0.13 a | 1.82 ± 0.18 a | 6.58 ± 0.35 b | 5.63 ± 0.41 a | 1.54 b | 1.22 b |
| Tomato | 0.49 ± 0.44 b | 1.45 ± 0.41 a | 1.72 ± 0.26 c | 2.08 ± 0.25 b | 1.92 ± 0.26 a | 1.52 ± 0.28 b | 8.19 ± 0.50 a | 6.00 ± 0.60 a | 2.98 a | 1.94 a |
| Chinese cabbage | 1.44 ± 0.18 a | 1.34 ± 0.16 a | 2.12 ± 0.57 b | 2.68 ± 0.24 b | 0.73 ± 0.13 b c | 0.71 ± 0.09 c | 2.64 ± 0.18 c | 1.88 ± 0.14 c | 1.27 b | 0.76 c |
| Annual | 1.97 ± 2.58 | 3.65 ± 1.10 | 8.01 ± 3.71 | 10.69 ± 2.22 | 4.62 ± 0.52 | 4.05 ± 0.54 | 17.41 ± 1.03 | 13.50 ± 1.15 | 1.94 | 1.35 |

All values were presented with Mean ± SE, *n* = 6.

**Table 3.** Annual $CH_4$ and $N_2O$ emissions, vegetable yield, net SGWP and GHGI as affected by biochar and N fertilizer.

| Treatments | $CH_4$ | $N_2O$ | Yield | SGWP | GHGI |
|---|---|---|---|---|---|
| | kg $CH_4$ ha$^{-1}$ | kg $N_2O$-N ha$^{-1}$ | t ha$^{-1}$ | kg $CO_2$-eq ha$^{-1}$ yr$^{-1}$ | kg $CO_2$-eq per yield yr$^{-1}$ |
| Control | 1.97 ± 2.58 | 4.62 ± 0.52 | 181.14 ± 4.13 | 2146 ± 335 | 11.84 ± 3.21 |
| Biochar (B) | 3.65 ± 1.10 | 4.05 ± 0.54 | 185.47 ± 4.12 | 1881 ± 276 | 10.14 ± 2.69 |
| N fertilizer (N) | 8.01 ± 3.71 | 17.41 ± 1.03 | 200.16 ± 3.46 | 7745 ± 601 | 38.69 ± 7.36 |
| B+N | 10.69 ± 2.22 | 13.50 ± 1.15 | 201.88 ± 2.47 | 6208 ± 588 | 30.75 ± 2.28 |
| Biochar (B) | * | ** | NS | ** | * |
| N fertilizer (N) | ** | *** | ** | *** | *** |
| Interaction (B × N) | NS | ** | NS | * | NS |
| Model | * | *** | *** | *** | ** |

\*, \*\*, and \*\*\* indicate statistically significant at the 0.05, 0.01, and 0.001 probability levels by a two-way ANOVA, respectively; NS, not significant. All values were presented with Mean ± SE, *n* = 6. SGWP-net flux-sustained global warming potentials; GHGI-the greenhouse gas intensity.

Over the annual cropping rotation, soil $CH_4$ fluxes were consistently and positively related to soil water content across all treatments (Figure 2a,b).

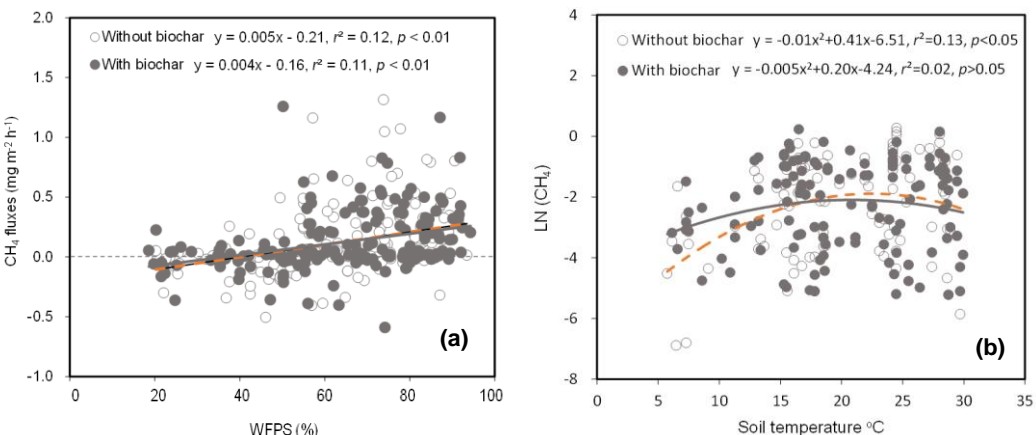

**Figure 2.** Dependence of $CH_4$ fluxes on soil WFPS over the annual greenhouse vegetable cropping system(**a**); Dependence of $CH_4$ fluxes on soil temperature (**b**), $CH_4$ fluxes was log-transformed, LN means natural logarithm.

### 3.2. Soil $N_2O$ Fluxes

Similarly, annual dynamics of soil $N_2O$ fluxes differed greatly in magnitude, but not in the direction across all treatments, depending largely on N fertilization and irrigation events (Figure 1d). Soil $N_2O$ fluxes were observed mainly occurring during the vegetable-growing seasons, while relatively lower in the inter-cropping fallow periods. The peaks of $N_2O$ fluxes were mainly measured within one week following basal fertilization and subsequent topdressing events, especially when accompanied by irrigation practice. In regard of with and/or without biochar amendment, soil $N_2O$ fluxes averaged 198.2 µg $N_2O$-N m$^{-2}$ h$^{-1}$ and 254.6 µg $N_2O$-N m$^{-2}$ h$^{-1}$ for the N fertilized treatments, respectively. Annual soil $N_2O$ emissions depended significantly on N fertilizer, biochar and their interaction according to the results of two-way ANOVA (Table 3).

Biochar significantly decreased the annual soil release rate of $N_2O$ by 22% and 12% for the N fertilized and unfertilized plots, respectively. This mitigation effect of biochar was further enhanced with fertilizer N combination (Table 3). On average, biochar combined with N fertilizer application reduced fertilizer N-induced soil $N_2O$ emissions by 31% relative to N applied alone (1.94%). The regression slope of N fertilizer presence relative to its absence was significantly lower than the value when soil $N_2O$ fluxes in biochar-treated soils were plotted against those without biochar amendment (Figure 3a: slope = 0.74,

$p < 0.001$), indicating that the mitigation capacity of $N_2O$ resulting from biochar addition would be enhanced with N fertilizer input in the vegetable cropping system.

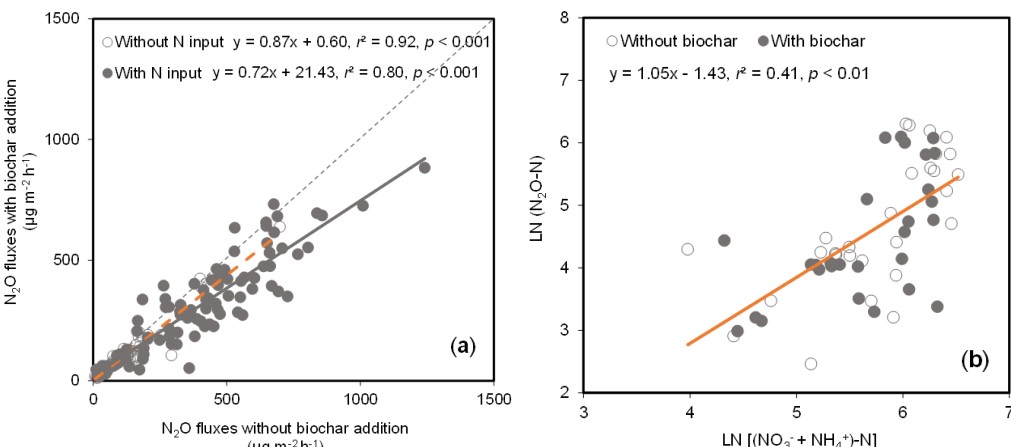

**Figure 3.** Soil release of $N_2O$ fluxes (**a**) in biochar-amended treatments against those without biochar-amended treatments and its dependence on soil mineral N contents ($NO_3^--N+NH_4^+-N$, **b**). Both soil $N_2O$-N fluxes and mineral N contents were all log-transformed. The thin dotted line represents the 1:1 line where the values from biochar-amended and unamended soils are equal, while the solid and dotted lines in bold represent linear regressions for individual observations. LN means natural logarithm.

### 3.3. Vegetable Yield

Vegetable yields in fresh weight were increased by N fertilizer application (Table 3). Highest vegetable yield as a sum of the three consecutive vegetable crops occurred for B+N treatment, which was 11% greater than that in control treatment. Fertilizer application significantly enhanced vegetable yields with or without biochar presence (Table 3). However, biochar effect alone and its interactive effect with N fertilizer on yield were not pronounced across the annual rotation cycle.

### 3.4. Net SGWP and GHGI

In order to evaluate the climatic impact of greenhouse vegetable cropping systems derived from $CH_4$ and $N_2O$ emissions under biochar amendment, we calculated the combined SGWP of $CH_4$ and $N_2O$. The SGWP presented a range from 1.88 t $CO_2$-eq ha$^{-1}$ year$^{-1}$ to 7.75 t $CO_2$-eq ha$^{-1}$ year$^{-1}$ when averaged over all experimental treatments (Table 3). Overall, an identical positive net SGWP across treatments in this study makes the vegetable soil a net source of GHGs. The net SGWP of vegetable systems was significantly affected by biochar, N fertilizer and their interaction (Table 3). Fertilizer N applied alone significantly increased net SGWP by 260% as compared with the control. Biochar amendment on the contary significantly decreased the net SGWP, and to a greater extent with N fertilizer combination.

The annual GHGI relating the net SGWP to vegetable yield was also significantly affected by N fertilizer and biochar, but not their interaction (Table 3). The GHGI was significantly lower in biochar-treated plots than in control plots and this decrease was further enhanced when N fertilizer was applied. Compared with the control, N fertilizer applied alone significantly enhanced the GHGI by 226%. In contrast, biochar decreased the GHGI of vegetable systems by 20% when combined with N fertilizer input over the annual vegetable cropping rotation.

## 4. Discussion

### 4.1. Roles of N Fertilizer and Biochar in Regulating Soil $CH_4$ and $N_2O$ Emissions

Methane is primarily formed under oxygen-deprived soil conditions where methanogenic archaea can utilize soil C input by vegetation growth or organic materials addition as their ultimate source of organic substrates [25–27]. However, soil production of $CH_4$ could be simultaneously consumed by methanotrophs in soils. Therefore, the net soil release or uptake of $CH_4$ depends on the balance resulting from combined performance of both methanogenic and methanotrophic communities. Recently, biochar as a potential soil improver and carbon sequestrator has been increasingly reported to affect greenhouse gas emissions in agricultural soils, generally showing a decrease [28] or no effect [29]. The open field vegetable soils were generally documented as net sink of atmospheric $CH_4$ due to the rapid loss of irrigation water. However, greenhouse vegetable soils can also act as a weak source of atmospheric $CH_4$, especially following frequent irrigation with enriched soil C substrate (Tables 2 and 3, Figure 1b,c). Biochar amendment significantly increased $CH_4$ emissions, largely attributed to the biochar-induced soil labile C enrichment available to methanogenic microbes [20]. Besides the biochar-induced soil labile C input, the seasonal intensive irrigation events in this study also showed the formation of anaerobic soil conditions driving $CH_4$ production. However, N-fertilizer greatly weakened the positive response of $CH_4$ to biochar, suggesting that the presence of N might have changed to direct soil labile C loss largely through crop growth and thus reduce its availability to other soil microbial processes [30].

In contrast, N fertilizer application significantly increased $CH_4$ emissions, particularly when biochar was also applied (Table 3, Figure 1c), suggesting that biochar-introduced C into the soil, improving soil C/N ratio towards enhancing activities of methanogens. Nevertheless, the repsonse of soil $CH_4$ fluxes to N fertilization has been inconsistent among earlier studies [10,31–33], a huge dependency on the N fertilizer type or rate, soil moisture and other soil properties has been reported [34–37]. In this study, the N fertilizer-increased soil $CH_4$ emissions may be presumably associated with the stimulated crop growth, and in turn more root exudated C substrate that was provided for methanogenesis resulting from bochar amendment [38]. However, no significant interactive effect of biochar with N fertilizer has been found on $CH_4$ emissions over the annual experimental period (Table 3).

In general, the production of $N_2O$ in soil is well documented through the microbially-mediated soil processes of nitrification and denitrification [39,40]. However, soil $N_2O$ generation or consumption processes are highly subject to soil moisture, temperature and mineral N contents [10,41,42]. As shown in Figure 1a,d and Figure 3b, soil $N_2O$ fluxes generally peaked after N fertilization events and increased with soil WFPS and soil mineral N contents. Similar to the seasonal pattern of $CH_4$, soil $N_2O$ fluxes were mainly measured during vegetable growing seasons relative to those during the intercropping fallow periods (Figure 1d). Therefore, N fertilization, soil moisture, soil mineral N and crop growth might have together accounted for the seasonal variation of $N_2O$ fluxes [8,43,44].

Despite the fact that soil $N_2O$ emissions had high inter-seasonal variations, annual total $N_2O$ emissions in this study was comparable to previous reports in intensive vegetable cropping systems in subtropical China [45]. Biochar significantly reduced soil $N_2O$ emissions in comparison to unamended soils in this study, and the mitigation potential was enhanced with N fertilizer combination (Table 3, Figures 1e and 3a). Nonetheless, the extent of biochar to mitigate soil $N_2O$ emissions was observed to significantly differ among individual vegetable crops, showing the largest mitigation potential in cabbage and tomato growing seasons with and without N fertilizer presence, respectively (Table 2). Previously, the potential of biochar to decrease soil $N_2O$ emissions has gained much attention [10,19,22,46], largely attributed to soil microbial N immobilization caused by biochar amendment. Presumably, the greater mitigation effect when in combination with N fertilizer input suggested that biochar-induced soil $N_2O$ mitigation potential might have been improved in vegetable soils with high background N levels. In particular, the amendment of biochar with high C/N ratio might suppress soil N turnover rate and thus result in the

decline of soil N availability for $N_2O$ production [47]. In addition, the observed mitigation effect of biochar on $N_2O$ may also be associated with the biochar-enhanced nitrogen use efficiency (NUE) as previously reported in vegetable systems, particularly in N-enriched arable soils [48].

*4.2. Direct Emission Factor and Background Emission of $N_2O$*

In this study, the annual emission factors of N fertilizer for $N_2O$ were estimated to be 1.35% (0.76–1.94%) and 1.94% (1.27–2.98%) with and without biochar amendment, respectively, but varied significantly among individual vegetable crops (Table 2). Our findings were generally greater than those of 0.63% obtained in southeast China and those of 0.24–0.30% in northeast China [8,13], while comparable to estimates of 1.1–1.9% in a similar Chinese greenhouse vegetable system [49]. In general, the emission factors of $N_2O$ in the present study were within the range of annual emission factors (0.41–5.0%) for conventional vegetable fields in China [9,49,50]. However, a recent meta-analysis pointed out that greenhouse and open-field vegetable systems had similar N fertilizer-induced emission factor of $N_2O$, but the results were only based on a limited data [43]. Over the annual cycle, soil $N_2O$ emissions were significantly affected by the interaction of N fertilizer with biochar, suggesting that the extent of biochar to mitigate $N_2O$ emissions depended on the presence or absence of N fertilizer (Table 3). On average, biochar amendment decreased the annual emission factor of $N_2O$ by 30% relative to unamended soils, suggesting that biochar could reduce fertilizer N-induced direct $N_2O$ emissions in vegetable cropping systems.

Soil background emissions, defined as $N_2O$ emissions from soils without N fertilizer input or other soil amendments attribution, have been recognized as important components for national agricultural $N_2O$ budget [51]. On average, annual background emission of $N_2O$ was estimated to be 4.62 kg N ha$^{-1}$ year$^{-1}$ for this study, significantly greater than the value of 2.67–2.87 in Chinese greenhouse [49]. Also, our results are higher than those (1.22–2.0 kg N ha$^{-1}$ year$^{-1}$) for the grain croplands [4,52]. Presumably, the high background emission of $N_2O$ in this study was mainly due to the frequent irrigation and intensive tillage events. On the other hand, the residual N from preceding vegetable cropping years due to excessive N inputs might have also contributed to this high emission rate. Besides, long-term excessive residual N retention in this study resulted in high nitrate contents in the background soils (51.1–365.9 mg N kg$^{-1}$ soil), as supported by the positive relationship of $N_2O$ fluxes with soil mineral N contents (Figure 3b).

*4.3. Net SGWP and GHGI Response to Biochar and N Fertilizer*

A trade-off relationship between $CH_4$ and $N_2O$ emissions was often observed in soils following biochar amendment, such as a stimulation of $CH_4$ emissions, but a suppression of $N_2O$ emissions [53]. Therefore, we need a comprehensive assessment in terms of their combined effects on climate change. In the present study, we calculated the annual net combined SGWP (t $CO_2$ equivalent ha$^{-1}$) of $CH_4$ and $N_2O$ to evaluate the climatic impact of greenhouse vegetable systems as affected by N fertilization and biochar amendment. Annual net SGWP was positive for all treatments, suggesting that vegetable soil acted as a net source of atmospheric GHGs [10,45]. In particular, N fertilization largely intensified this source role, especially in soils with no biochar amendment (Table 3). In addition, the GHGI relating the SGWP to vegetable yield was also significantly decreased following biochar amendment, while this mitigation effect was enhanced with N fertilizer combination (Table 3). Therefore, the source role of greenhouse vegetable cropping systems in shaping climate change could be largely altered by biochar amendment, particularly in vegetable soils with high N availability.

**5. Conclusions**

Overuse of N fertilizer and frequent irrigation in greenhouse vegetable soils have gained much attention in China. This study provided an insight into the interactive role of biochar with N fertilizer affecting soil GHGs fluxes in a vegetable cropping system. Biochar

amendment significantly increased soil $CH_4$ emissions, while N-fertilizer decreased soil $N_2O$ emissions, irrespective of N fertilizer application. N-fertilizer addition weakened the positive response of $CH_4$ to biochar while enhancing the biochar-induced $N_2O$ mitigation potential. Biochar amended relative to unamended soils decreased N fertilizer-induced annual direct $N_2O$ emissions in greenhouse vegetable systems. The SGWP and GHGI were significantly decreased by biochar amendment but this mitigation was enhanced by N fertilizer combination. Overall, this research shows that biochar amendment may mitigate the climatic impact of greenhouse vegetable systems but without reducing vegetable yield in subtropical China, particularly when amended to soils with high N enrichment levels.

**Author Contributions:** Conceptualization, Y.Z., J.W. and L.Y.; methodology, Y.Z. and Y.Y.; software, Y.Z.; validation, Y.Z.; formal analysis, Y.Z.; investigation, Y.Z.; resources, Y.Z. and A.H.; data curation, Y.Z.; writing—original draft preparation, Y.Z.; writing—review and editing, D.H., J.W. and L.Y.; visualization, Y.Z.; supervision, J.W. and L.Y.; project administration, Y.Y. and J.W.; funding acquisition, J.W. and L.Y. All authors have read and agreed to the published version of the manuscript.

**Funding:** This research was funded by Jiangsu Agricultural Science and Technology Innovation Fund, grant number CX(21)1002; Huai'an Natural Science Research Project (HABZ201917).

**Data Availability Statement:** Not applicable.

**Conflicts of Interest:** The authors declare no conflict of interest.

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
