# Peer review of "Biochar-Induced Mitigation Potential of Greenhouse Gas Emissions Was Enhanced under High Soil Nitrogen Availability in Intensively-Irrigated Vegetable Cropping Systems"

_agronomy, doi:10.3390/agronomy12102249_

Round 1
Reviewer 1 Report
The study investigated the influence of biochar and N fertilization on soil CH4 and N2O emissions in a vegetable cropping system and found that biochar amendment significantly increased soil CH4 emissions by 33% and 85%, while it decreased 21 soil N2O emissions by 22% and 12% with and without N fertilizer input; Fertilizer N 22 combination weakened the positive response of CH4 to biochar while enhanced the mitigation 23 potential of biochar for N2O. However, the manuscript needs further revision. Discussion needs to be further updated and references should be more recent (currently, the latest paper was before 2019)
In line 17, what is the difference of agricultural system and vegetable system? You stated that the interactive effects of biochar with soil N availability in vegetable systems have yet to be clarified, have you checked with similar studies? For example, Li et al (2015) https://doi.org/10.1016/j.atmosenv.2014.10.034 , this topic has been extensively studied. Please state the novelty of your study.
In line 79, you stated the experiment is a field experiment while in line 97, it is in a greenhouse field?
In line 86, did you make the biochar or buy from the market, why select biochar that produced at 350-550 degree C, low temperature biochar may induce more GHG emissions.
Line 109, 40 t/ha rate is quite high, why this amount of biochar addition increased CH4 emssions? Have you checked with similar studies?
Line 280 did you measure the labile C in the biochar you are using? Even though the biochar may have high labile C, while it is still relative low compare to the portion of labile C to SOC.
In discussion, did you consider the adsorption capacity of biochar on available N?
Reviewer 2 Report
Dear authors,
Thank you for the opportunity to read your MS. I hope this feedback will find you with great spirit to improve the MS. The authors have an interesting experiment and showed information about the “Biochar-induced mitigation potential of greenhouse gas emissions was enhanced under high soil nitrogen availability in intensively-irrigated vegetable cropping systems”. Please, find below some suggestions I raised about your MS.
Title could change to “N2O” and not “Greenhouse gases”, because the results demonstrated a increase of CH4 emission that is also a Greenhouse gas
Lines 64-65: Add some suggestions about how the biochar interacts with N fertilizer to influence the net flux-sustained global warming potential (SGWP) and greenhouse gas intensity (GHGI) of CH4 and N2O.
Lines 74-75: There is no need of this information “We assumed that the overall climatic impact of vegetable cropping systems derived from CH4 and N2O emissions would be decreased by biochar amendment and N fertilizer presence would alter this mitigation effect.”
In the Material and Methods, give more information about the cultivation of vegetable annual rotation consisting of capsicum (Capsicum annuum), tomato (Solanum lycopersicum) and Chinese cabbage (Brassica chinensis)
The crops were planted in the same vase?
Rate of N. Explain the high N rates. Why the rate of 40 t ha-1 of biochar? Which N source? Add more information about fertilizer managements
How Biochar was added in soil? Volume of vases?
Add information of intensive irrigation events
When soil was collected?
Information of Figure 1 is not clear. I can see the difference between the treatments. Maybe, the authors could separate the Figures into more Figures.
Add an analysis (ANOVA) in the results of Table 1. I can identify if the treatments were different based on ANOVA.
How to interpretate the result of ANOVA in Tables 2 and 3. The authors must add the ANOVA letters.
The authors could calculate the N lost based on N rate, because the rate of N applied in the study is high. Provably, this effect will be more intense. Another idea is comparing the emission of N2O with the time of application in each. There is different crops and the results, probably is different.
